# *Soybean Mosaic Virus* 6K1 Interactors Screening and GmPR4 and GmBI1 Function Characterization

**DOI:** 10.3390/ijms24065304

**Published:** 2023-03-10

**Authors:** Ting Hu, Hexiang Luan, Liqun Wang, Rui Ren, Lei Sun, Jinlong Yin, Hui Liu, Tongtong Jin, Bowen Li, Kai Li, Haijian Zhi

**Affiliations:** 1National Center for Soybean Improvement, Nanjing Agricultural University, Nanjing 210095, China; 2Institute of Plant Genetic Engineering, College of Life Science, Qingdao Agricultural University, Qingdao 266109, China; 3College of Agronomy, Henan Agricultural University, Zhengzhou 450046, China

**Keywords:** *Soybean mosaic virus*, 6K1, interaction, pathogenesis-related 4, bax inhibitor 1

## Abstract

Host proteins are essential during virus infection, and viral factors must target numerous host factors to complete their infectious cycle. The mature 6K1 protein of potyviruses is required for viral replication in plants. However, the interaction between 6K1 and host factors is poorly understood. The present study aims to identify the host interacting proteins of 6K1. Here, the 6K1 of *Soybean mosaic virus* (SMV) was used as the bait to screen a soybean cDNA library to gain insights about the interaction between 6K1 and host proteins. One hundred and twenty-seven 6K1 interactors were preliminarily identified, and they were classified into six groups, including defense-related, transport-related, metabolism-related, DNA binding, unknown, and membrane-related proteins. Then, thirty-nine proteins were cloned and merged into a prey vector to verify the interaction with 6K1, and thirty-three of these proteins were confirmed to interact with 6K1 by yeast two-hybrid (Y2H) assay. Of the thirty-three proteins, soybean pathogenesis-related protein 4 (GmPR4) and Bax inhibitor 1 (GmBI1) were chosen for further study. Their interactions with 6K1 were also confirmed by bimolecular fluorescence complementation (BiFC) assay. Subcellular localization showed that GmPR4 was localized to the cytoplasm and endoplasmic reticulum (ER), and GmBI1 was located in the ER. Moreover, both *GmPR4* and *GmBI1* were induced by SMV infection, ethylene and ER stress. The transient overexpression of GmPR4 and GmBI1 reduced SMV accumulation in tobacco, suggesting their involvement in the resistance to SMV. These results would contribute to exploring the mode of action of 6K1 in viral replication and improve our knowledge of the role of PR4 and BI1 in SMV response.

## 1. Introduction

Soybean (*Glycine max* (L.) Merr) is one of the most important commercial and cereal crops cultivated on a large scale in the world. However, *Soybean mosaic virus* (SMV) seriously harms soybean yield and seed quality [1,2]. SMV is a member of the potyviruses, which are composed of many agriculturally and economically important viruses, such as *Turnip mosaic virus* (TuMV), *Plum pox virus* (PPV), *Tobacco etch virus* (TEV), and *Potato virus Y* (PVY), which cause serious losses in crop production [3,4]. SMV infection usually results in leaf mosaic, wrinkling, necrosis, and plant dwarfing in the infected plants, and can cause significant losses—as high as 86% of soybean yield [5]. Based on the differential symptoms of a series of soybean cultivars, SMV isolations were divided into twenty-two SMV strains (SC1–SC22) in China, and seven SMV strains (G1–G7) in the United States [6,7,8].

Similar to other potyviruses, the SMV genome consists of single-stranded, positive-sense RNA molecules of approximately 10 kilobases (kb) and encodes eleven mature multifunctional proteins, namely P1, HC-Pro, P3, P3N-PIPO, 6K1, CI, 6K2, VPg, NIa-Pro, NIb, and CP [9,10]. The functions of most potyviral proteins have been reported. The CP, HC-Pro, CI, 6K2, VPg, and P3N-PIPO proteins are involved in viral movement [11,12,13,14,15,16,17]. Moreover, P3, CI, 6K2, VPg, NIa-Pro, and NIb proteins participate in viral replication [18,19,20,21,22,23], and Hc-Pro, P3 and CI proteins play roles in potyviral virulence and pathogenicity [24,25,26,27]. Among the eleven potyviral proteins, 6K1 was first detected and purified in vivo in 2006 from *Plum pox virus* (PPV)-infected *N. benthamiana;* 6K1 and 6K2 are the two smallest proteins with similar molecular weights of 6 kDa, but little attention has been paid to 6K1 compared to 6K2 [28]. Then, the SMV 6K1 protein was reported to localize to the periphery of infected leaf cells of *Pinellia ternata*, suggesting the possible role of 6K1 in viral cell-to-cell movement [29]. Subsequent studies have shown that the mature 6K1 protein is required for potyviral replication and is an essential element of the viral replication complex (VRC) at the early infection stage [30]. Furthermore, it was reported that the 6K2 protein of *Tobacco vein banding mosaic virus* (TVBMV; genus *Potyvirus*) interacted with 6K1 and recruited 6K1 to VRC to regulate potyvirus replication [31]. However, the precise function of 6K1 in viral replication remains unclear.

The synthesis, folding and maturation of proteins occur in the endoplasmic reticulum (ER). When plants are under abiotic and biotic stresses, the nascent and unfolded proteins can rapidly accumulate in the ER and exceed its folding capacity, resulting in ER stress and activating the unfolded protein response (UPR) reaction that promotes protein folding and protects plants from stresses [32]. Plant virus (e.g., potexviruses and potyviruses) infection can induce ER stress. The UPR pathway restricts *Plantago asiatica mosaic virus* (PlAMV; genus *Potexvirus*) and *Turnip mosaic virus* (TuMV; genus *Potyvirus*) accumulation in plants, mainly through enhancing the protein folding capacity of the ER [33]. SMV infection causes ER stress, which in turn promotes virus accumulation in soybean [34].

Plant hormones, such as salicylic (SA), jasmonic (JA) and ethylene (ET), play significant roles in regulating plant defense to potyviruses. The SA signal pathway leading to the induction of system acquired resistance (SAR) is essential in the resistance to viral, fungal, and bacterial pathogens [35,36]. AGD2-like defense response protein 1 (NbALD1) induced resistance to *Turnip mosaic virus* (TuMV) via positively regulating the SA pathway and negatively regulating the ET pathway in *N. benthamiana* [37]. ET signaling mutants reduced *cauliflower mosaic virus* (CaMV) accumulation via restricting viral long-distance movement [38]. The induction of the ET pathway is important for successful TuMV infection in Arabidopsis [39]. The JA pathway mainly protects plants against necrotrophic pathogens and wounding [40]. In addition, potyviruses infection could alter the phytohormone signal. For example, HC-Pro enhances the expression of defense-related genes in the SA pathway, and 6K1 can inhibit the JA-dependent defenses [41,42]. Exploring the response of host factors to ER stress and phytohormones would facilitate the clarification of the molecular mechanism of host resistance to potyviruses.

The potyviral genome is relatively small with limited coding capacity, and viral proteins must interact with numerous host factors to establish the systemic infection of host plants [43,44]. Here, we screened the soybean cDNA library using SMV-6K1 as the bait to search for host factors involved in SMV replication, and the protein–protein interaction (PPI) networks were constructed. Then, two 6K1 interactors, GmPR4 and GmBI1, were further studied to analyze their role in the resistance to SMV. This work could lead to a better understanding of the interaction between 6K1 and host proteins during infection, and serve to develop effective strategies for crop protection against potyviruses.

## 2. Results

### 2.1. 6K1 Localizes to Nucleus, ER and Cytoplasm

To find conserved motifs of 6K1 in different SMV isolates, the 6K1 sequences of 30 SMV isolates were retrieved from the NCBI database and subjected to multiple sequence alignment. SMV-6K1 is 52 amino acids in length, and the 6K1 proteins from different SMV isolates are highly conserved (Figure 1A). To examine the intracellular distribution of 6K1 in planta, the 6K1 of SMV strain SC15 was fused with green fluorescent protein (GFP) and transiently expressed in *N. benthamiana* leaf epidermis cells. The results showed that 6K1 was present in the cytoplasm, ER and nucleus (Figure 1B).

### 2.2. Evaluation of the pBT3-STE-6K1 Bait Vector

The 6K1 of SMV strain SC15 was fused with pBT3-STE to form the bait plasmid pBT3-STE-6K1. Then, the auto-activation activity of pBT3-STE-6K1 was tested. Yeast cells transformed with pBT3-STE-6K1 could grow on SD/-Leu agar plates but did not grow on SD/-Leu/-His/-Ade agar plates. Yeast cells co-transformed with pBT3-STE-6K1 and pAI-Alg5 produced colonies on SD/-Leu/-Trp/-His/-Ade agar plates, while yeast cells co-transformed with pBT3-STE-6K1 and pPR3-N did not grow on SD/-Leu/-His/-Ade agar plates, indicating no toxicity and auto-activation of 6K1 (Figure 2). Therefore, the pBT3-STE-6K1 construct is suitable for cDNA library screening.

### 2.3. Screening of Soybean Proteins Interacting with SMV-6K1

To identify the potential host proteins interacting with 6K1, pBT3-STE-6K1 was used as the bait to screen the soybean cDNA library. The yeast colonies grew well on the SD/-Leu/-Trp/-His/-Ade agar plates were collected and yeast plasmids were extracted for PCR detection using sequencing primer pPR3-N-F/R. Only the plasmids with a single band detected by PCR were considered to be positive, and 252 positive colonies were selected for sequencing. The sequencing results were analyzed using the Soybase database. Finally, we obtained 127 potential soybean interactors of 6K1 (Appendix A). According to protein function annotations, these proteins were classified into six categories, namely defense-related, transport-related, DNA binding, metabolism-related, unknown, and membrane-related proteins (Figure 3). Defense-related proteins accounted for the largest percentage (28%), including Bax inhibitor 1 (BI1), pathogenesis-related protein 1 (PR1), Chitinase class I (PR3), pathogenesis-related protein 4 (PR4), papain family cysteine protease (PFCP), and cysteine protease inhibitors (CYSB). This was followed by transport-related proteins (20%), such as triose-phosphate transporter family (TPT), ZIP zinc transporter (ZIP), and major intrinsic protein family (MIP). Then, metabolism-related, DNA binding, unknown, and member-related proteins accounted for 17%, 17%, 11% and 7%, respectively.

To further confirm the interaction, 39 soybean proteins from the 127 potential 6K1 interactors were cloned and merged into the prey vector pPR3-N (Table 1). Then, the 39 recombinant prey vectors were each co-transformed with pBT3-STE-6K1 into yeast cells to verify the interactions with 6K1. Among the 39 proteins, 33 proteins produced colonies on SD/-Leu/-Trp/-His/-Ade agar plates, which were confirmed to interact with 6K1 (Appendix A).

### 2.4. Protein-Protein Interaction Networks of SMV-6K1 Interactors

In order to analyze the interaction relationship among the 6K1 interactors, the homologues of the 127 soybean proteins in *A. thaliana* were collected, and then 95 Arabidopsis proteins were obtained (Appendix A). The Gene Ontology (GO) analysis and protein–protein interaction (PPI) networks of the 95 Arabidopsis proteins were performed using the STRING database (95 items (Arabidopsis thaliana)—STRING interaction networks (string-db.org)). The GO analysis showed that the transport, transmembrane transporter activity, cytoplasm and membrane were the most enriched terms under the biological process, molecular function, and cellular component categories (Figure 4A). Forty-eight proteins connecting with other proteins were shown in this networks, and they were generally grouped into protein synthesis and transport-related, photosynthesis-related, and defense-related proteins, according to the function description of corresponding proteins (Figure 4B). We note that the 47 proteins with no connection to other proteins were not shown in the networks.

### 2.5. Subcellular Localization and BiFC Analyses of GmPR4 and GmBI1

GmPR4 (*Glyma.03g247500*) and GmBI1 (*Glyma.05g064700*), belonging to defense-related proteins in the PPI networks with strong interactions with 6K1, were selected for further study. To examine the localization of GmPR4 and GmBI1 in planta, they were merged with green fluorescent protein (GFP) and transiently expressed in *N. benthamiana* leaf epidermis cells. Results showed that PR4 was located in theER and cytoplasm, and BI1 was localized in the ER (Figure 5A). The interactions between 6K1 and PR4/BI1 were further confirmed by bimolecular fluorescence complementation (BiFC) assay. YFP fluorescence signals were observed with 6K1-YC and PR4/BI1-YN combinations, and no fluorescence signals were detected in the negative control (Figure 5B). Combining the results of the yeast two-hybrid (Y2H) and BiFC assays, GmPR4 and GmBI1 interacted with 6K1.

### 2.6. Expression Analyses of GmPR4 and GmBI1 by qRT-PCR

The responses of *GmPR4* and *GmBI1* to SMV infection were analyzed by qRT-PCR (Figure 6). The results showed that *GmPR4* was induced in both resistant and susceptible soybean cultivars. The expression levels of *GmPR4* in Nannong 1138-2 (SMV-susceptible) showed obvious up- and down-regulation patterns before and after 6 h post inoculation (hpi), respectively, with a maximum expression of approximately eightfold at 6 hpi, suggesting that *GmPR4* rapidly responded to SMV at the early infection stage in Nannong 1138-2. In Kefeng No.1 (SMV-resistant), *GmPR4* expression levels were up-regulated from 6 to 72 hpi, with a maximum expression of approximately sixfold at 24 hpi (Figure 6A). As for *GmBI1*, the expression patterns induced by SMV were opposite in resistant and susceptible soybean cultivars. *GmBI1* expression levels were obviously up-regulated in Kefeng No.1 but down-regulated in Nannong 1138-2 (Figure 6B). These results suggested the involvement of PR4 and BI1 in the response to SMV infection in soybean.

Moreover, in order to analyze the responses of *GmPR4* and *GmBI1* to phytohormones and ER stress, their expression levels were examined under phytohormones (salicylic (SA), jasmonic (JA) and ethylene (ET)) and ER stress treatments in Nannong 1138-2. ET is a gaseous hormone, and 1-Aminocyclopropane-1-carboxylic acid (ACC), the precursor of ET, was used as the ET-inducing agent. Dithiothreitol (DTT) was used as the ER stress-inducing agent, as it disrupts disulfide bonding and affects protein folding, leading to ER stress [45]. Results showed that *GmPR4* was up-regulated from 4 to 12 h post treatment (hpt) with ACC treatment, with a maximum expression of approximately ninefold at 9 hpt, and *GmPR4* was also up-regulated by approximately sixfold at 12 hpt with DTT (Figure 6C), suggesting the involvement of *GmPR4* in ET and ER stress pathways. *GmBI1* expression levels remained relatively stable by SA treatment and were up-regulated by approximately threefold at 9 hpt with JA and ACC treatments. We note that *GmBI1* was significantly up-regulated from 2 to 12 hpt under DTT treatment, with a maximum expression of approximately sixfold at 4 hpt (Figure 6D), indicating that *GmBI1* was obviously involved in the ER stress pathway.

### 2.7. Transient Overexpression of GmPR4 and GmBI1 Reduced SMV Accumulation in N. benthamiana

To examine how GmPR4 and GmBI1 responded to SMV infection, we transiently co-expressed GmPR4 and GmBI1, respectively, with the infectious clone pCB301-SMV-SC7::GFP in *N. benthamiana*. GmPR4 and GmBI1 were inserted into a pGD vector to form the recombinant plasmids pGD-PR4 and pGD-BI1. Then, the agrobacterial cultures containing empty vectors pGD, pGD-PR4 and pGD-BI1 were co-infiltrated with the infectious clone pCB301-SMV-SC7::GFP into the *N. benthamiana*, respectively. The expression levels of PR4 and BI1 were examined by qRT-PCR at 0, 24, 48, and 72 h post infiltration, and results showed that PR4 and BI1 were overexpressed effectively in *N. benthamiana* (Appendix A). The accumulation of SMV in the infiltrated leaves was detected by ELISA and qRT-PCR at 7 days post infiltration. Results showed that there was obviously less SMV accumulation in the transient overexpression plants of PR4 and BI1 than the empty vector-expressed plants (Figure 7A,B). Next, we observed the GFP signals in the upper leaves of *N. benthamiana*, which reflected the content of SMV. Compared with empty vector-expressed plants, there were less GFP signals in the transient overexpression plants of PR4 and BI1 (Figure 7C). These results indicated that GmPR4 and GmBI1 positively regulated the resistance to SMV in *N. benthamiana*.

## 3. Discussion

The potyviral 6K1 protein is involved in viral replication and is an essential element of the 6K2-induced viral replication complex (VRC), which consists of viral proteins (e.g., 6K2, NIb, P3, and CI) and host factors, for robust viral replication [20,30]. However, not much work has been devoted to characterizing the precise function and molecular mechanism of 6K1 in viral replication. In this study, we screened the soybean proteins interacting with 6K1 by the yeast two-hybrid (Y2H) system. Interactions detected in the Y2H system cannot mimic natural interactions between the virus and plant during infection. Therefore, the potential interactions found in the Y2H system may not happen during natural infection. However, we have observed the interactions between 6K1 and the Papain family cysteine protease (GmPFCP)/Cysteine protease inhibitors (GmCYSB) that were reported previously [42], suggesting that the Y2H system is a good system to analyze interaction in vitro. The identification of host factors interacting with 6K1 contributes to the understanding of the molecular mechanism of 6K1 in virus infection.

Plant virus genomes are relatively small with a very limited coding power, and the complex interactions between host factors and viral proteins are necessary for plant viruses to generate successful infection [44]. Previous studies have identified many host factors interacting with potyviral proteins, including P3, 6K2, VPG, and NIa-Pro [46,47,48]. Here, 127 soybean proteins were preliminarily identified to interact with SMV-6K1 based on the DUALmembrane system. According to their function description, these proteins were divided into six categories, namely defense-related, transport-related, metabolism-related, DNA binding, unknown, and membrane-related proteins (Figure 3). In this categorization, defense-related proteins accounted for the most (28%), followed by transport-related proteins (20%), suggesting the involvement of 6K1 in viral movement, which is consistent with previous reports [29,42]. This suggestion was further supported by GO analysis, which revealed that candidate genes were significantly enriched in the transport process and transmembrane transporter activity (Figure 4A). To explore the relationship among 6K1 interactors and search for significant genes, the PPI networks were constructed (Figure 4B). In these networks, the proteins involved in protein synthesis and transport formed a large networks containing 18 proteins, and AT5G59240, AT5G12110, ELF5A-1, RPS29A, and RPS13A were found to be hub genes by MCODE analysis. These are the soybean homologous genes of 40S ribosomal protein S8 (GmRPS8), elongation factor 1-beta (GmEF1), translation initiation factor 5A (GmEIF5A), 40S ribosomal protein S29 (GmRPS29), and 40S ribosomal protein S13 (GmRPS13), respectively, and their interactions with 6K1 may promote viral replication.

Several issues should be taken into consideration regarding these 6K1 interactors. First, 6K1 was partially localized in the nucleus, and the interactions occurring in the nucleus may not be detected since the screening of the soybean cDNA library was based on the DUALmembrane system. Second, the gene fragments screened by 6K1 are not complete gene coding sequences. Some of the identified proteins may result from spurious interactions, which need to be further confirmed by other methods (e.g., Y2H, BiFC or Co-IP). Therefore, the coding sequences of 39 candidate genes were cloned and fused into the prey vector pPR3-N to confirm the interactions with 6K1, and 33 proteins of them were verified as positive interactors (Table 1 and Appendix A). Interestingly, of the six negative interactors, five proteins were predicted to be partially localized in the nucleus (Table 1), suggesting that the identification of interactions by the Y2H experiment based on the GAL4 nuclear system may be necessary. As a result, GmRPS8-1 and GmRPS8-2, two of the six negative interactors, were identified to interact with 6K1 by the Y2H experiment of GAL4 system. Among the 33 positive interactors, GmBI1 and GmPR4 were selected for further study on account of both of them being defense-related proteins that appeared in the PPI networks with strong interactions with 6K1 (Appendix A and Figure 4B), and previous studies have shown their potential role in the defense against potyvirus (see below).

PR4 was reported to respond to pathogen attack and abiotic stress in plants. CaPR4 is a defense-related gene conferring resistance against TMV in pepper via the JA and ET pathway [49]. OsPR4 genes are involved in abiotic stress responses and tolerance, in addition to their responsiveness to *Magnaporthe grisea* infection in rice [50]. The knockdown of VvPR4 increased grape susceptibility to downy mildew of grapevine (*Vitis vinifera* L.) [51]. Our results showed that *GmPR4* was up-regulated in both susceptible and resistant soybean cultivars after SMV infection, suggesting the involvement of GmPR4 in the defense against SMV (Figure 6A). In addition, *GmPR4* was strongly induced by ACC treatment, implying that GmPR4 may play a role in the ET pathway (Figure 6C). Except the role in the ripening of fruits or leaf abscission, ET also plays an active role in plant defense [52,53,54]. Furthermore, the transient overexpression of GmPR4 reduced SMV accumulation in *N. benthamiana* (Figure 7), and these results indicate that GmPR4 may participate in the resistance to SMV through the ET pathway.

Previous studies have reported the roles of BI1 in biotic and abiotic stress. Bax inhibitor 1 (BI1) is a conserved ER protein, suppressing cell death in animals and plants and regulating a cell death pathway, which is important for cell preservation during ER stress [55,56,57,58,59]. AtBI1 was rapidly up-regulated during wounding and pathogen challenge and restricted *Plantago asiatica mosaic virus* (PlAMV) and *Turnip mosaic virus* (TuMV) infection in *A. thaliana* [60,61,62]. Moreover, some reports have shown that BI1 negatively regulated the resistance to several pathogens in plants. The expression of AtBI1 strongly reduced the resistance to the brown rust fungus in transgenic sugarcane plants [58]. Transgenic barley expressing HvBI1 was more susceptible to powdery mildew but less susceptible to *F. graminearum* [63]. Our studies showed that GmBI1 was localized to the ER, which is consistent with previous reports [64,65]. Expression analysis showed that *GmBI1* was up-regulated in the resistant soybean cultivar but down-regulated in the susceptible soybean cultivar (Figure 6B). The reason might be that the SMV accumulation inhibited the expression levels of GmBI1 in the susceptible soybean cultivar. *GmBI1* was also induced by JA and ACC treatments and strongly up-regulated by DTT treatment (Figure 6D), and the transient overexpression of GmBI1 decreased the SMV accumulation in *N. benthamiana* (Figure 7). These results suggest that GmBI1 may be involved in the resistance to SMV, mainly through the ER stress pathway.

In this study, we screened the host proteins interacting with 6K1 and obtained 127 potential 6K1 interactors. Next, functional classification and GO analysis were performed, which suggested the role of 6K1 in viral movement (Figure 3 and Figure 4A). Although the involvement of 6K1 in viral movement has been also proposed in previous studies [29,42], more direct evidence is required. Moreover, two 6K1 interactors, PR4 and BI1, were further studied. We preliminarily demonstrated the involvement of PR4 and BI1 in the resistance to SMV by a transient overexpression experiment in tobacco. Subsequent work focusing on the precise functions and molecular mechanisms of PR4 and BI1 in SMV infection in soybean is necessary. The further functional characterization of other 6K1 interactors is also worthwhile, which could improve the understanding of the molecular mechanisms of virus infection and provide potential molecular targets for the genetic improvement of a resistant soybean cultivar.

## 4. Materials and Methods

### 4.1. Plant Materials, Virus Strains and Soybean cDNA Library

The soybean cultivars Kefeng No.1 (SMV-SC3-resistant) and Nannong 1138-2 (SMV-SC3-susceptible) were grown in a greenhouse at 25 °C in the day and 20 °C at night, with a photoperiod of 16 h. The *N. benthamiana* plants were maintained in a greenhouse with 16 h of light at 23 °C and 8 h of darkness at 20 °C. The SMV strains SC3 and SC15 were used in this study. The soybean cDNA library was prepared from the leaves of Nannong 1138-2 infected by an SMV strain SC15 [47]. All materials were provided by the National Center for Soybean Improvement (NCSI), Nanjing Agricultural University (NJAU), Nanjing, China.

### 4.2. Gene Cloning, Vector Construction and Primers

Total RNA was extracted from the soybean leaves of SC15-infected Nannong 1138-2 using TRIzol reagent (Vazyme, Nanjing, China) and was used to synthesize the first-strand cDNA using the Prime Script RT Reagent Kit (TaKaRa, Dalian, China). The full-length coding sequences of soybean genes and SMV-6K1 were cloned from the cDNA using specific primers based on the sequences of target genes and were purified by agarose gel recovery.

For 6K1 bait vector construction, the coding sequence of 6K1 was cloned using primer pair sfi-6K1-F/R. Next, both 6K1 and pBT3-STE were digested by the Sfi I (NEB, Beijing, China) enzyme and connected by the T4 DNA ligase (TaKaRa, Dalian, China) to generate the bait plasmid pBT3-STE-6K1. The target genes of soybean were individually cloned into the prey vector pPR3-N using Gateway technology [66]. The vectors used in BiFC and localization assays were also constructed by Gateway technology. In order to develop the plasmids for subcellular localization, the coding sequences of target genes were connected into the pGWB6 plasmid, respectively. For vectors used in BiFC assay, the interaction proteins were introduced into the Gateway vectors pEarleyGate202-YN and pEarleyGate201-YC, respectively. To construct the transient overexpression vectors, the pGD plasmid was digested with SmaI and BamHI (NEB, Beijing, China). Then, the linearized vector and gene fragments were ligated together by the homologous recombination method to generate the pGD-PR4 and pGD-BI1 plasmids. All the recombinant plasmids were confirmed by DNA sequencing, and all primers used in this study are listed in Appendix A.

### 4.3. Bioinformatic Analysis

The 6K1 amino acid sequences of 30 SMV strains were obtained using the NCBI database (https://www.ncbi.nlm.nih.gov (accessed on 14 August 2022)) and were aligned by the DANMAN software. Subcellular localization prediction was performed by the WoLF PSORT server (https://wolfpsort.hgc.jp (accessed on 21 December 2022)). The homologues of 127 soybean proteins in *A. thaliana* were searched using the TAIR website (https://www.arabidopsis.org/ (accessed on 26 February 2022)), and only 95 Arabidopsis proteins were obtained for the following reasons. Some soybean proteins belong to the same gene family and correspond to the same homologous protein in Arabidopsis. Moreover, several soybean proteins have no homologous protein in Arabidopsis. The Gene Ontology (GO) analysis and protein–protein interaction (PPI) networks of the 95 Arabidopsis proteins were performed using the STRING database (https://cn.string-db.org (accessed on 22 February 2022)). The PPI networks obtained from the STRING database were processed with Cytoscape software. MCODE, a plugin in Cytoscape, was used to screen hub genes in the PPI networks.

### 4.4. Yeast Two-Hybrid (Y2H) Assay

The screening of the soybean cDNA library by Y2H assay was carried out using the DUALmembrane kit 3 (Dualsystems Biotech) according to the manufacturer’s protocols, and the yeast strain NMY51 was used in study. pAI-Alg5, provided by DUALmembrane kit 3, was used as the positive control prey. The pBT3-STE-6K1 plasmid was used as the bait to screen the soybean cDNA library. Yeast plasmids of positive colonies that grew well on SD/-Leu/-Trp/-His/-Ade agar plates were extracted using E.Z.N.A. Yeast Plasmid Kit and transformed into DH5α competent cells. Next, the DH5α cells containing yeast plasmids were grown on LB medium with ampicillin of 50 μg/mL and detected by PCR using primer pair pPR3-N-F/R. Only plasmids with a single band detected by PCR were chosen for sequencing, the sequencing results were analyzed using the Soybase database (SoyBase.org). The interactions between 6K1 and soybean proteins were further confirmed by Y2H assay. The coding sequences of target genes were cloned and fused with pPR3-N using the Gateway technology to generate the prey vectors. Next, the bait vector pBT3-STE-6K1 and prey vectors were co-transformed into the yeast cells (NMY51), grown on SD/-Leu/-Trp and SD/-Leu/-Trp/-Ade/-His agar plates, and incubated at 30 °C for 3–4 days.

### 4.5. Localization and Bimolecular Fluorescence Complementation (BiFC) Assays

The recombinant plasmids used for subcellular localization were introduced into the *A.tumefaciens* strain EHA105 via electroporation. Agrobacterial cultures contained relevant expression vectors were grown overnight on YEB medium with kanamycin and rifampicin at 28 °C, and then the cultures were pelleted by centrifugation and subsequently resuspended to an optical density of 0.8 at 600 nm (OD_600_) in infiltration buffer (10 mM MgCl_2_ [pH 5.6], 10 mM MES, and 150 μM acetosyringone). Next, the cell suspensions were agroinfiltrated into one-month-old *N. benthamiana* plants and the fluorescence signal was visualized under a confocal microscope. RFP containing the ER-retention peptide HDEL was used as the ER marker (ER-RFP) [67]. H_2_B-mCherry was used as the nucleus marker. For the BiFC assay, the interaction proteins were fused with pEarleyGate202-YN/pEarleyGate201-YC and transformed into EHA105 cells. A mixture of two EHA105 agrobacterial cultures containing relevant expression vectors were resuspended in infiltration buffer and infiltrated into *N. benthamiana*. The interaction between YN and 6K1-YC was used as the negative control. YFP expression was observed 48 h later under a confocal microscope. Subcellular localization and BiFC experiments were repeated at least three times.

### 4.6. Stress Treatments, Quantitative RT-PCR and ELISA

Soybean plant stress treatments were performed at the VC growth stage. The leaves of mock-inoculated (inoculated with PBS) and SC3-infected Nannong 1138-2 and Kefeng No.1 were collected at 0, 6, 12, 24, 48, 72, 96 and 120 h post inoculation (hpi). Nannong 1138-2 seedlings were sprayed with SA (200 μM SA), JA (100 μM MeJA), ET (100 μM ACC) and DTT (10 mM DTT) in 0.1% (*v*/*v*) ethanol and 0.1% (*v*/*v*) silwet L-77, respectively. For the mock control, Nannong 1138-2 seedlings were treated with 0.1% (*v*/*v*) ethanol and 0.1% (*v*/*v*) silwet L-77. Then, the leaves were harvested at 0, 2, 4, 6, 9, 12, and 24 h post treatment (hpt). Each sample was independently collected and stored at −80 °C immediately after freezing in liquid nitrogen. The total RNA of soybean leaves was extracted using TRIzol reagent (Vazyme, Nanjing, China), and first-strand cDNA was synthesized using HiScript^®^ II QRT SuperMix (Vazyme, Nanjing, China) following the manufacturer’s protocol. Quantitative RT-PCR (qRT-PCR) was performed with a CFX96 real-time PCR detection system (Bio-Rad, Hercules, CA, USA) using 2x SYBR Green Master Mix (Vazyme, Nanjing, China) according to the manufacturers’ instructions. The soybean gene *Tubulin* (accession No. AY907703) was employed as an internal control to normalize the cDNA. Transcript levels were quantified by the relative quantification (2^−ΔΔCt^) method. Each sample was tested in three biological replicates and the same experiment was performed three times. ELISA was conducted with an antibody diagnostic kit (V094-R2, Nanodiaincs, Fayetteville, NC, USA) following the manufacturer’s protocol. The absorbance value of samples at 405 nm was tested by Infinite 200PRO (TECAN, Männedorf, Switzerland). The significant differences were calculated by one-way ANOVA (*p* < 0.05).

### 4.7. Agrobacterium-Mediated Transient Overexpression

The infectious clone of pCB301-SMV-SC7::GFP was constructed and confirmed to systemically infect *N. benthamiana* [27,68]. The gene expression plasmids (pGD, pGD-PR4 and pGD-BI1) and pCB301-SMV-SC7:GFP were transferred into EHA105 cells. The agrobacterial culture mixtures of gene expression plasmid and infectious clone were resuspended to an optical density of 1 at 600 nm (OD_600_) in infiltration buffer at a ratio of 500:1, then agro-infiltrated into one-month-old *N. benthamiana*. The expression levels of PR4 and BI1 were examined by qRT-PCR at 0, 24, 48, 60, and 72 h post infiltration, and the *β-Tubulin* gene (accession No. U91564) of tobacco was used as the internal control to normalize the cDNA. The infiltrated leaves were collected at 7 days post infiltration, and the SMV accumulation was detected by ELISA and qRT-PCR. The GFP fluorescence in the upper non-infiltrated leaves was observed under ultraviolet light at 12 days post infiltration.

## Figures and Tables

**Figure 1 ijms-24-05304-f001:**
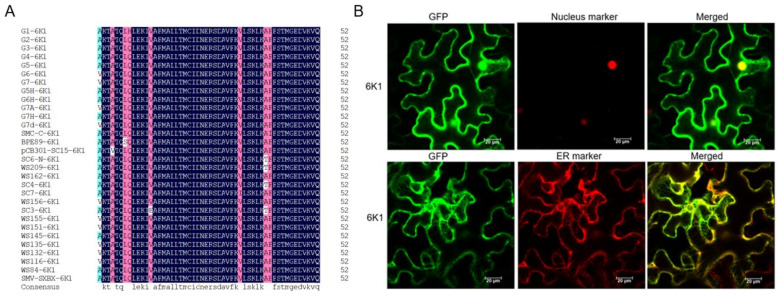
Sequence alignment and subcellular localization of SMV-6K1. (**A**) A multiple sequence alignment of 6K1 protein sequences of 30 SMV isolates. The sequences of 30 SMV isolates were retrieved from the NCBI database and the accession numbers are shown: G1 (FJ640977.1), G2 (S42280.1), G3 (FJ640978.1), G4 (FJ640979.1), G5 (AY294044.1), G6 (FJ640980.1), G7 (AF241739.1), G5H (FJ807701.1), G6H (FJ640981.1), G7A (FJ640982.1), G7H (FJ807700.1), G7d (AY216987.1), SMV-C (LC323107.1), BPE89 (MW655827.1), pCB301-SC15 (MH919386.1), SC6-N (KP710867.1), WS209 (FJ640976.1), WS162 (FJ640973.1), SC4 (MN539670.1), SC7 (MH919385.1), WS156 (FJ640971.1), SC3 (MH919384.1), WS155 (FJ640970.1), WS151 (FJ640969.1), WS145 (FJ640967.1), WS135 (FJ640965.1), WS132 (FJ640964.1), WS116 (FJ640961.1), WS84 (FJ640956.1), SMV-SXBX (MT712111.1). The conserved amino acid residues among the 30 proteins are shaded in dark blue. (**B**) Subcellular localization of SMV-6K1. The 6K1-GFP fusion protein was transiently co-expressed with nucleus marker and endoplasmic reticulum (ER) marker in *N. benthamiana* leaf epidermis cells, individually. The fluorescence signals were observed by confocal microscopy 2 days post agroinfiltration. Scale bars = 20 μm.

**Figure 2 ijms-24-05304-f002:**
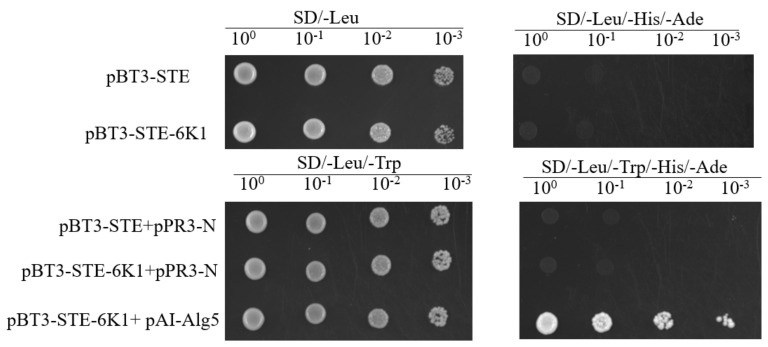
Evaluation of the pBT3-STE-6K1 bait vector. Yeast cells co-transformed with pBT3-STE-6K1 and pAI-Alg5 were use as the positive control, and pBT3-STE and pPR3-N were used as the negative control. The yeast grown cells were diluted by 1-, 10-, 100-, and 1000-fold.

**Figure 3 ijms-24-05304-f003:**
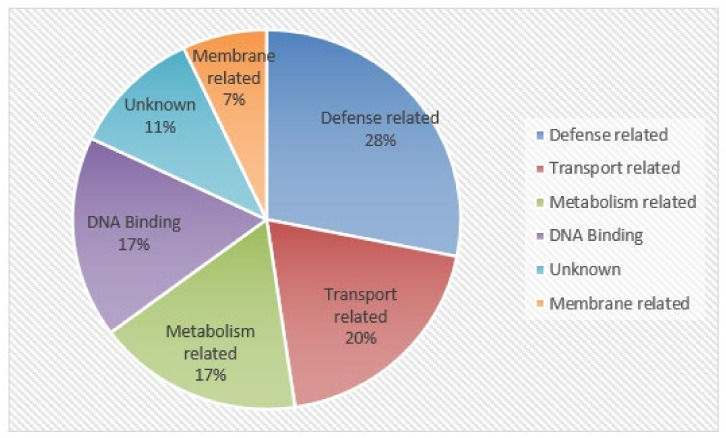
Function classification of potential soybean proteins interacting with 6K1 based on their putative function.

**Figure 4 ijms-24-05304-f004:**
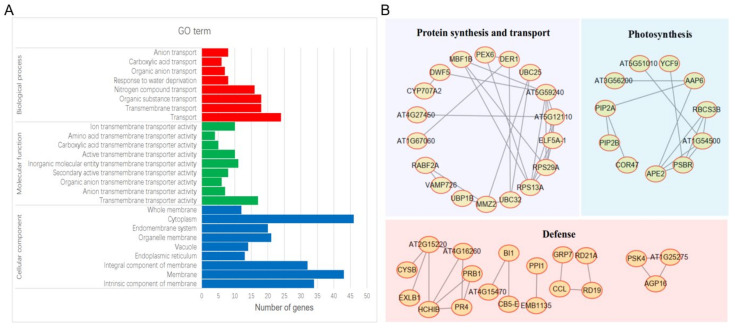
Analysis of homologues of 6K1 interactors in *A. thaliana* using STRING database. Gene Ontology (GO) analysis (**A**) and protein–protein interaction networks (**B**) of homologues of 6K1 interactors in *A. thaliana*.

**Figure 5 ijms-24-05304-f005:**
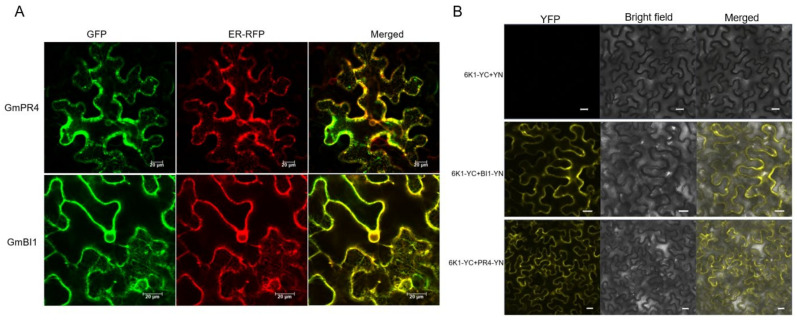
Subcellular localization of GmPR4/GmBI1 and analysis of interactions between 6K1 and PR4/BI1 by BiFC assay. (**A**) Subcellular localization of PR4 and BI1 in *N.benthamiana* leaf epidermis cells. (**B**) Verifying the interactions between 6K1 and PR4/BI1 by BiFC assay. PR4/BI1-YN and 6K1-YC were co-agroinfiltrated into leaves of one-month-old *N. benthamiana*. Interaction between 6K1-YC and YN was used as the negative control. Scale bars = 20 μm.

**Figure 6 ijms-24-05304-f006:**
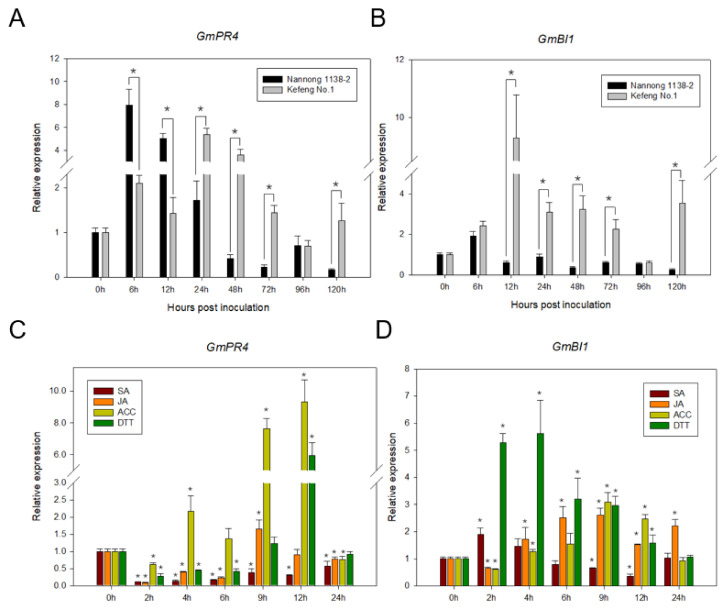
Expression analyses of *GmPR4* and *GmBI1* by qRT-PCR. The relative expression of *GmPR4* (**A**) and GmBI1 (**B**) in Nannong 1138-2 (SMV-susceptible) and Kefeng No.1 (SMV-resistant) after SMV infection at 0, 6, 12, 24, 48, 72, 96, and 120 hpi. Y-axes indicate the ratios of relative expression levels between samples infected with SMV and samples inoculated with PBS (phosphate buffer saline), and X-axes indicate the hours post inoculation. The significant differences between susceptible and resistant plants were tested by one-way ANOVA, * *p* < 0.05. The relative expression of *GmPR4* (**C**) and *GmBI1* (**D**) after various stress treatments (SA, JA, ACC, and DTT) at 0, 2, 4, 6, 9, 12, and 24 hpt. Y-axes indicate the ratios of relative expression levels between samples under various stress treatments (SA, JA, ACC and DTT) and control treatment, X-axes indicate the hours post treatment. Each experiment was repeated three times. Error bars indicate standard deviation (SD). The significant differences were tested by one-way ANOVA and asterisks denote significance relative to 0 h for each treatment, *p* < 0.05. The soybean gene *Tubulin* was used as the internal control to normalize the cDNA.

**Figure 7 ijms-24-05304-f007:**
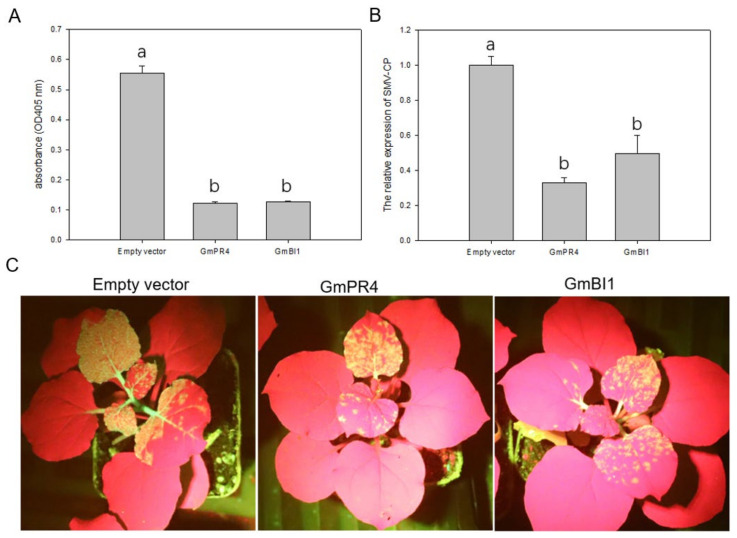
Transient overexpression of GmPR4 and GmBI1 in *N. benthamiana.* The accumulation of SMV in infiltrated leaves was detected by ELISA (**A**) and qRT-PCR (**B**) at 7 days post infiltration. Each experiment was repeated three times with error bars indicating standard deviation (SD). The significant differences were tested by one-way ANOVA, *p* < 0.05. Bars with identical letters are not significantly different. (**C**) The GFP signals in upper leaves were observed under ultraviolet light at 12 days post infiltration.

**Table 1 ijms-24-05304-t001:** Identification of proteins interacting with SMV-6K1.

Protein Name	Gene	Function Description	Function Classification	Identification of Interaction ^1^	Subcellular Location ^2^
GmBI1-1	*Glyma.01g205200*	Inhibitor of apoptosis-promoting Bax1	Defense-related	+	ER
GmPR3	*Glyma.02G042500*	Chitin recognition protein, PR3	Defense-related	+	Extra
GmPR4	*Glyma.03g247500*	Barwin family, chitinase-related	Defense-related	+	Extra
GmDFP	*Glyma.04g009400*	Dehydrin family protein	Defense-related	+	N
GmPFCP-1	*Glyma.04g027600*	Papain family cysteine protease	Defense-related	+	Extra
GmBI1–2	*Glyma.05g064700*	Inhibitor of apoptosis-promoting Bax1	Defense-related	+	ER
GmMT2A	*Glyma.07g132000*	Metallothionein	Defense-related	+	Ch, N
GmERG4	*Glyma.08g224200*	Ergosterol biosynthesis ERG4/ERG24 family	Defense-related	+	PM
GmTLP	*Glyma.10g061000*	Thaumatin family	Defense-related	+	Extra
GmBI1-3	*Glyma.11g037700*	Inhibitor of apoptosis-promoting Bax1	Defense-related	+	ER
GmELP	*Glyma.11g154900*	Extensin-like protein repeat	Defense-related	+	Extra
GmALIP	*Glyma.12g150500*	Aluminium induced protein	Defense-related	−	Cyto, N, Ch
GmMBF1	*Glyma.12g129100*	Multiprotein bridging factor 1	Defense-related	+	N, Mito
GmCYSB-1	*Glyma.13g071800*	Cysteine protease inhibitors	Defense-related	+	Cyto
GmBI1-4	*Glyma.13g211300*	Inhibitor of apoptosis-promoting Bax1	Defense-related	+	ER
GmCYSB-2	*Glyma.13g189500*	Cysteine protease inhibitors	Defense-related	+	V
GmPFCP-2	*Glyma.14g085800*	Papain family cysteine protease	Defense-related	+	Cyto, N
GmPR1	*Glyma.15g062400*	Cysteine-rich secretory protein family	Defense-related	+	Extra
GmPSK	*Glyma.17g233500*	Phytosulfokine precursor protein	Defense-related	+	Ch, Extra
GmTPT1	*Glyma.07g201300*	Triose-phosphate Transporter family	Transport-related	+	Ch
GmZIP	*Glyma.08g328000*	ZIP Zinc transporter	Transport-related	+	PM
GmTPT2	*Glyma.13g175100*	Triose-phosphate Transporter family	Transport-related	+	Ch, Mito
GmMIP-1	*Glyma.13g325900*	Major intrinsic protein	Transport-related	+	PM
GmMIP-2	*Glyma.19G181300*	Major intrinsic protein	Transport-related	+	PM
GmRD	*Glyma.03G245100*	Rubredoxin	Metabolism-related	+	Ch, N
GmPSBR-1	*Glyma.08g173700*	Photosystem II 10 kDa polypeptide PsbR	Metabolism-related	+	Ch
GmP4Hc	*Glyma.11g080700*	Prolyl 4-hydroxylase alpha subunit	Metabolism-related	+	Ch
GmPSBR-2	*Glyma.15g253700*	Photosystem II 10 kDa polypeptide PsbR	Metabolism-related	+	Ch
GmUBC	*Glyma.17g032800*	Ubiquitin-conjugating enzyme	Metabolism-related	+	G
GmRPS8–1	*Glyma.03G086400*	40S ribosomal protein S8	DNA Binding	−	Ch, N
GmRRM	*Glyma.11g117600*	RNA recognition protein	DNA Binding	−	N
GmEF1	*Glyma.13g073200*	Elongation factor 1 beta/delta chain	DNA binding	−	Cyto, N
GmRPL18A	*Glyma.13g261500*	60S ribosomal protein L18A	DNA Binding	+	PM
GmRPL34	*Glyma.13g209500*	60S ribosomal protein L34	DNA Binding	+	Mito
GmRPS8-2	*Glyma.16g087700*	40S ribosomal protein S8	DNA Binding	−	Cyto, N
GmEIF5A	*Glyma.17g103100*	Translation initiation factor 5A (eIF-5A)	DNA Binding	+	Cyto
GmUP1	*Glyma.04g111000*	Unknown protein	Unknown	+	Ch
GmUP2	*Glyma.08g183000*	Unknown protein	Unknown	+	Extra, Ch
GmPMP	*Glyma.04G167300*	Predicted membrane protein	Membrane-related	−	PM

^1^ + means positive interaction with 6K1, − means negative interaction with 6K1. ^2^ ER, endoplasmic reticulum; Extra, extracellular; Ch, chloroplast; N, nucleus; PM, plasma membrane; Mito, mitochondrion; Cyto, cytoplasmic; V, vacuolar; G, golgi.

## Data Availability

All data are provided in the manuscript and its Appendix A.

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
