# Peer review of "Soybean Mosaic Virus 6K1 Interactors Screening and GmPR4 and GmBI1 Function Characterization"

_ijms, 2023, doi:10.3390/ijms24065304_

Round 1

Reviewer 1 Report

This manuscript describes the interaction network of soybean mosaic virus 6K protein. Confirmed yeast two-hybrid 6K1 cellular partners were classified according to their function, and the most abundant group was those of defence related proteins. Two 6K1 partners from this group, GmPR4 and GmBI1, were further characterized and showed to be SMV inducible and involved in the resistance to SMV. Those results contribute to deciphering the SMV defence response mechanism and the role of 6K1 in SMV pathogenesis. The comments for improvement are provided below.

Major comments:

Line 87: “The results showed that 6K1 was present in cytoplasm and nucleus, but not in ER (Figure 1B)”. To suggest nuclear localization of 6K1, co-localization with a nuclear marker is indispensable. ER marker does not look convincing. Upon confocal microscopy, ER normally looks like a network inside the cell (eg: Lazareva, E.A., Lezzhov, A.A., Chergintsev, D.A., Golyshev, S.A., Dolja, V.V., Morozov, S.Y., Heinlein, M. and Solovyev, A.G. (2021), Reticulon-like properties of a plant virus-encoded movement protein. New Phytol, 229: 1052-1066. https://doi.org/10.1111/nph.16905). This is not the case here; we can see only the borders of the cell and there are no visible ER like structures. Also, the nature of the ER-RFP marker is not revealed neither in the results nor in the materials and methods sections. Moreover, on the Fig 5A bottom panel, it even looks like ER-RFP localizes to the nucleus. Overall, quality of confocal microscopy figures is not suitable for publication.

Line 219: “pBin-GFP, pBin-PR4 and pBin-BI1 were co-infiltrated with the infectious clone pCB301-SMV-SC7::GFP into the N. benthamiana, respectively.” Co-expression of pBin-GFP with pCB301-SMV-SC7::GFP can result in stronger antiviral RNAi and reduce virus titre in control. A more suitable negative control would be empty pBin.

Minor comments

The part of introduction dedicated to defence response (lines 55-71) is, to my opinion, rather subdued and complicated for the reader. I would suggest making it more structured for a greater understanding.

Line 81: “2.1.6. K1 Localizes to the Nucleus and Cytoplasm”. Remove dot after 6.

Line 106: “And yeast cells co-transformed with pBT3-STE-6K1 and pAI- Alg5 produced colonies on SD/-Leu/-Trp/-His/-Ade agar plates.” What is Alg5? Why was it used as a positive control? Please include more details and a citation.

Line 146: “In order to analysis the interaction relationship among the 6K1 interactors”. Replace by “in order to analyse”. 

Fig5B: missing information on the top of the panel

Line 277: “Therefor”. Replace by “therefore”.

Reviewer 2 Report

Dear Authors,

I have an opportunity to review paper entitled : “Soybean Mosaic Virus 6K1 Interactors Screening and GmPR4 & GmBI1 Function Characterization” submitted to IJMS;

Authors concentrated on interaction between 6K1 of SMV and host proteins. Interactors were classified into six groups, including defense related, transport related, metabolism related, DNA binding, unknown and membrane related proteins. Moreover, soybean pathogenesis-related protein 4 (GmPR4) and Bax inhibitor 1 (GmBI1) were chosen for further study. Furthermore, their interactions with 6K1 were also confirmed by bimolecular fluorescence complementation (BiFC) assay. What was very interesting and potential overexpression of GmPR4 and GmBI1 reduced SMV accumulation in tobacco, suggesting their involvement in the resistance to SMV.

Please, underline deeply the aim of the study;

Secondly, Please underline that the subcellular localization was done only in epidermis tissue;

Moreover, I suggest to adjust A and B panel of Figure 5 to the “one figure”- in current version it looks quite strange;

The proteins screening  - I consider it the greatest advantage of this work and finding interactors of course;

Despite of the sentence “GmPR4 (Glyma.03g247500) and GmBI1 (Glyma.05g064700), belonging to defense related proteins in the PPI networks with strong interactions with 6K1 were selected for 164 further study.” I suggest that it should be deep explanation why exactly these two proteins from defense were selected (e.g. why PR4 );

Please, inform about reference genes in figure captions, not only in M&M section; Is it only one -tubulin reference gene ?

Because discussion chapter is rather not expanded, I suggest to add future prospects coming from Author’s obtained results, to make more visible and even more important these results for wider audience;

Sincerely

Reviewer 3 Report

The study of Hu et al investigates the function of 6K1 protein, a small protein encoded by potyviruses. Till now, not much is known about the 6K1 protein. It is a preliminary study but an impactful contribution to the field of potyvirus research however, there is a scope to improve the current manuscript which is mentioned below.

- The authors have claimed very less research has been done on 6K1 protein which is true. However, authors have failed to cite the work done on 6k1 protein published recently in "Viruses" (Bera et al., 2022., https://doi.org/10.3390/v14061341). I enjoyed reading the current Hu et al manuscript as it confirms some of the previous findings related to jasmonic acid induction, cysteine protease inhibitors, and 6K1's role in virus movement as mentioned in Bera et al., 2022. I would kindly request the authors to update the discussion with proper citations which actually makes their story strong. Studying interactions in yeast2hybrid system does not mimicks natural virus infection in plants. So, the potential interactions authors found in the Y2H system may not happen during natural infection. However, authors have observed several interactions that were reported previously in Viruses. This suggests that the Y2H system might be a good system to study 6K1 interactions in vitro.

- The introduction has lots of scopes to improve to let readers know about the 6K1 protein's history which is difficult to work with. Some relevant citations which first purified the 6k1 protein and characterized its location were from SMV also need to be cited here. doi: 10.1007/s00705-007-0972-7. ; https://doi.org/10.1099/vir.0.81873-0;  doi: 10.1038/srep43455.

- line 55 to line 71: Most of the positive-strand Plant RNA viruses replicate in cytoplasm and not in the cytoplasm. Citation number 29 which the authors have referred does not talks about virus replication. I would kindly request the authors to cite the correct papers. Overall this paragraph needs to be re-written as it is not clear here why or how phytohormones and ER stress are connected. Again, the number 36 citation does not have any data on cauliflower mosaic virus. There are better recent works to show ET signalling in potyvirus infection e.g., Casteel et al., 2015. The role of JA pathway in potyvirus infection mediated by 6k1 protein is already known and should be cited here. Bera et al 2022

- line 87: wrong conclusion: Present in endoplasmic reticulum. yellow fluorescence is observed in the figure 1B. I would kindly request the authors to take a better picture that focuses on ER for example see figure check figure 6 in https://doi.org/10.1111/nph.15228 . Moreover, upon transient expression 6K1 protein is made in the ER so it should be present in ER also. When 6K1 protein is expressed during potyvirus infection, it won't locate in ER as viral translation is limited within cytoplasm. 

- Table 1:Please divide the table as per the function classification. Meaning proteins categorised into the same function coming together. It helps to follow up and helps the reader to go through easily after the figure 3.

- section 2.4: It is not clear why this analysis was done and how it goes with the current story? Already in the previous section authors have indicated the different functions. 

-line 147: Please explain why there is not 127 protein homologoues from Arabidopsis?

- line 191 to 192: what was the rationale of quantifying phytohormones?

- Fig 6c and D: in which cultivar? resistant or susceptible

- In most of the figures, significance is not indicated properly and number of biological replicates is not also mentioned.

- line 243 to 244: did authors check conservation for other potyviral proteins? within SMV it is reasonable that there is conservation in sequences in multiple potyviral proteins.

- Figure 5: It suggests proximity of 6K1 protein with PR4 and BI4 but not confirms the interactions. Co-immunoprecipitations need to be done for confirmation. Moreover, after the transient expression of the 6K1 protein if there is an induction of PR4 and BI4 that might be also good to support the presence of these interactions. Figure 6A and B after SMV infection shows induction of PR4 and BI4 but that can be due to other potyviral proteins also. Did the authors do any experiments to show 6K1 protein inhibits viral replication as it induces a defense response? 

Round 2

Reviewer 1 Report

After the first round of submission the manuscript was definitely improved but unfortunately all main issues remained unsolved. First of all quality of confocal microscopy figures is still not suitable for publication. To suggest nuclear localisation of protein, co-localisation with nuclear marker is mandatory. Although, it is normal that ER marker partially localise to the nuclear envelope it still should have an obvious ER localisation. On the figure 5A we can guess that structures close to the plasma membrane are possibly ER but it is not convincing. The same thing with the cytoplasmic localisation of 6K1 on the Fig. 1B - only periphery of the cells is visible. 

In the section 2.7 of the results it is definitely more relevant to have ectopically expressed GFP in all plants and not not only in the control. I still do not understand why authors were using GFP fusions of PR4 and BI1 and pBinGFP to co-express with GFP tagged virus. Transient expression of GFP increase antiviral RNAi and reciprocally GFP targeted virus increase RNAi of PR4 and BI1 constructs. Also it was not demonstrated that GFP fusions of PR4 and BI1 are functional. I would recommend to use untagged PR4 and BI1 (or tagged with different smaller tags e.g. HA or myc) constructs and empty pBin as a control, and if possible assess PR4 and BI1 accumulation levels.

Reviewer 3 Report

- I would kindly request the authors to incorporate their response number 8 in the manuscript under material and methods section which explains why there is not 127 protein homologoues from Arabidopsis?

- Fig 6 D could be better presented in the paper. Some of the error bars are missing at 9 and 12 hr. To show if the significance is different, above each bar significance should be indicated by an asterisk. In the legend, it should be mentioned asterisk denotes significance relative to 0 hr for each phytohormone treatment.

Round 3

Reviewer 1 Report

The authors improved the manuscript and addressed all my queries. I am happy to recommend it for publication 

Reviewer 3 Report

The authors have improved the paper and without any reservation, I would recommend it for publication.